# Multisite Injections of Canine Glial-Restricted Progenitors Promote Brain Myelination and Extend the Survival of Dysmyelinated Mice

**DOI:** 10.3390/ijms251910580

**Published:** 2024-10-01

**Authors:** Piotr Rogujski, Magdalena Gewartowska, Michal Fiedorowicz, Malgorzata Frontczak-Baniewicz, Joanna Sanford, Piotr Walczak, Miroslaw Janowski, Barbara Lukomska, Luiza Stanaszek

**Affiliations:** 1NeuroRepair Department, Mossakowski Medical Research Institute, Polish Academy of Sciences, 02-106 Warsaw, Poland; 2Electron Microscopy Research Unit, Mossakowski Medical Research Institute, Polish Academy of Sciences, 02-106 Warsaw, Poland; mgewartowska@imdik.pan.pl (M.G.); mbaniewicz@imdik.pan.pl (M.F.-B.); 3Small Animal Magnetic Resonance Imaging Laboratory, Mossakowski Medical Research Institute, Polish Academy of Sciences, 02-106 Warsaw, Poland; 4Sanford Biotech LLC, Tissue and Cells Bank, Sioux Falls, SD 57106, USA; joannasanford5@gmail.com; 5Program in Image Guided Neurointerventions, Department of Diagnostic Radiology and Nuclear Medicine, University of Maryland, Baltimore, MD 21201, USA; pwalczak@som.umaryland.edu (P.W.); miroslaw.janowski@som.umaryland.edu (M.J.)

**Keywords:** neurological disorder, demyelination, glial-restricted progenitor, cell therapy, multisite injection, myelination, survival, magnetic resonance imaging

## Abstract

Glial cell dysfunction results in myelin loss and leads to subsequent motor and cognitive deficits throughout the demyelinating disease course.Therefore, in various therapeutic approaches, significant attention has been directed toward glial-restricted progenitor (GRP) transplantation for myelin repair and remyelination, and numerous studies using exogenous GRP injection in rodent models of hypomyelinating diseases have been performed. Previously, we proposed the transplantation of canine glial-restricted progenitors (cGRPs) into the double-mutant immunodeficient, demyelinated neonatal shiverer mice (shiverer/Rag2^−/−^). The results of our previous study revealed the myelination of axons within the corpus callosum of transplanted animals; however, the extent of myelination and lifespan prolongation depended on the transplantation site (anterior vs. posterior). The goal of our present study was to optimize the therapeutic effect of cGRP transplantation by using a multisite injection protocol to achieve a broader dispersal of donor cells in the host and obtain better therapeutic results. Experimental analysis of cGRP graft recipients revealed a marked elevation in myelin basic protein (MBP) expression and prominent axonal myelination across the brains of shiverer mice. Interestingly, the proportion of galactosyl ceramidase (GalC) positive cells was similar between the brains of cGRP recipients and control mice, implying a natural propensity of exogenous cGRPs to generate mature, myelinating oligodendrocytes. Moreover, multisite injection of cGRPs improved mice survival as compared to non-transplanted animals.

## 1. Introduction

Neurological disorders are a major threat to public health [1,2,3]. Among all central nervous system (CNS) diseases, those of white matter are one of the most prevalent in neurology [4,5]. They include hereditary, congenital, or early myelin loss and acquired myelin disorders. Although the etiology and pathology of these diseases are most often known, the treatment is either lacking or restricted to minimalizing the symptoms [6,7]. Central white matter consists of myelin, oligodendrocytes, axons, and astrocytes. In the CNS, oligodendrocytes are responsible for myelin production, while astrocytes support the formation and maintenance of myelin [8,9,10]. Myelin is an essential part of the membrane sheath that protects the axon and helps maintain the balance of its internal environment. Its main function is to increase the speed and efficiency of nerve signal transmission in a process known as saltatory conduction. Myelin acts as an electrical insulator, enabling faster conduction of the nerve impulse, thus contributing to the proper functioning of sensory perception, motor skills, and various cognitive processes. Oligodendrocyte insufficiency or loss and deficits in the myelination of axons compromise the neurons’ survival and lead to motor dysfunction and disability [11,12]. The strategy to replace pathological cells or support their viability and function allows exogenous glial restricted progenitor (GRP) transplantation to serve as a potential therapeutic tool in demyelinating disorders [13,14]. Several experimental GRP replacement therapies have been performed in rodent models of hypo- and dysmyelinating diseases. The most commonly used model is the shiverer mouse, which has a deletion in the myelin basic protein (MBP) gene. Due to myelin deficiency, the homozygous shiverer mutant mice present a shivering phenotype, tonic seizures, and a shorter life (in our case, 28 weeks, on average) [15,16]. The pioneering studies of the Goldman group have proven that it is possible to obtain compact myelin and axonal myelination in the double-mutant (shiverer/rag2^/−^) mice by injecting human fetal GRPs in the forebrain subcortex of neonatal animals [17,18,19,20]. In these studies, glial-restricted progenitor transplant recipients revealed the donor-derived myelin, which ensheathed host axons in the brainstem. Moreover, multiple transplanted mice exhibited prolonged survival with neurological defect improvement as compared to non-treated shiverer mice.

The number of studies using non-human glial-restricted progenitors for transplantation in dysmyelinating diseases is relatively small. Interestingly, despite close similarities between GRPs originating from humans and other animal species in terms of cell phenotype, the effect of transplantation differed with regard to the distribution of donor cells in the host, the myelination process, and recipient survival [21,22]. In our previous studies, we used canine GRPs (cGRPs) isolated from the brains of dog fetuses and implanted these cells intraventricularly into the double-mutant, immunodeficient, dysmyelinated neonatal shiverer mice (shiverer/Rag2^−/−^). The selection of dogs as GRP donors has a practical advantage. There is a growing population of dogs as companion animals with neurological health issues requiring attention. One of the canine illnesses of the CNS is degenerative myelopathy (DM) [23,24]. Degenerative myelopathy affects mostly adult dogs and may be characterized by the loss of the upper motor neurons and degeneration of the nerve fibers [25]. It creates a solid market and demand for the application of cGRPs in veterinary medicine. Through the mapping of the cerebral bio-distribution of transplanted cGRPs in the mouse brain in our previous studies, the dispersion of donor cells was observed around the entire ventricular system. However, closer analysis confirmed that the most prominent density of transplanted cells was localized around the transplantation sites. MRI analysis revealed visible, although not statistically significant, myelination of the corpus callosum of cGRP-transplanted demyelinated animals. Myelination with compact myelin was also observed around some of the axons of transplanted animals by electron microscopy and in immunohistochemical staining. Nevertheless, the prolongation of the lifespan was only noticed when cells were transplanted posteriorly (near the ventricle lining the hippocampus) [26]. The goal of our present study was to optimize the therapeutic effect of cGRP transplantation by using a multisite injection protocol to achieve a broader dispersal of donor cells in the host and obtain better results.

## 2. Results

### 2.1. cGRP Phenotype

Canine GRPs grew in vitro as a monolayer and demonstrated typical glial progenitor features, including a bipolar morphology and adherence to poly-L-lysine/laminin-coated culture dishes. Over time, the cGRPs started forming compact, adherent cellular aggregates (Figure 1A). Phenotypically, at the third passage, cGRPs demonstrated the expression of specific glial progenitor/oligodendrocyte precursor markers, including A2B5, NG2, PDGFRα, Olig1, Olig2, O4, and CNPase. Approximately 17% of the cells revealed proliferation activity, as indicated by Ki67 expression. The cultured GRPs were negative for oligodendrocyte markers MBP or GalC as shown by immunocytochemistry (ICC) (Figure 1B).

In the third passage, most cGRPs remained in an undifferentiated, progenitor state, as indicated by flow cytometry analysis. The majority of cells expressed A2B5 and PSA-NCAM antigens (41.2% A2B5^+^/O4^−^; 50.2% A2B5^+^/PDGFRα^−^; 93.5% PSA-NCAM^+^/MBP^−^). There was only a small population of O4-positive cells (8.7% O4^+^/NG2^−^) and modest populations of GFAP^+^ (6%) and CD11b^+^ myeloid-lineage cells (4.4%). In detail: O4^+^/A2B5^−^ 4.2%, A2B5^+^/O4^+^ 4,2%, O4^+^/Ng^2+^ 0%; Ng^2+^/O4^−^ 0%; A2B5^−^/PDGFR^+^ 0.2%; A2B5^+^/PDGFR^+^ 0.6%; PSANCAM^+^/MBP^+^ 1.1%; MBP^+^/PSANCAM^−^ 0.1%. The content of mature, myelinating oligodendrocytes was very small (0.1% MBP^+^/PSANCAM^−^). Thus, the above results of flow cytometry analysis confirmed our immunocytochemical observations (Figure 2).

### 2.2. Immunohistochemical Analysis of Mouse Brains

Seven months after cGRP transplantation, host shiverer mouse brains were extensively myelinated both anteriorly (injected doubly) and posteriorly (injected triply), as indicated by global MBP expression. The highest distribution of MBP-positive signal was localized in the corpus callosum, hippocampus, and striatum. We also noticed, on average, 24% more MBP posteriorly, closer to the lambda-transplantation site (*p* = 0.0286). No MBP expression was detected across the brains of non-transplanted *Mbp*^−/−^ control mice (Figure 3).

Meanwhile, the proportion of cells expressing GalC within the cortex/corpus callosum and striatum was similar in cGRP-transplanted (17.72 ± 1.44% and 16.98 ± 2.06%, respectively) and control shiverer mice (13.79 ± 1.81% and 17.29 ± 2.40%, respectively) (*p* > 0.05) (Figure 4).

Mature oligodendrocytes were visible both in control shiverer mice and transplanted animals, as depicted by APC (CC1) staining in Figure 5.

No activated macrophages/microglia were detected in the brains of control and transplanted animals (Figure 6A,B). No glial scar was visible within the transplanted tissue, and no astrocyte activation was observed (Figure 6C–E).

### 2.3. Electron Microscopy Visualization of Myelin Presence in Mouse Brains

Ultrastructural visualization of the myelinated corpus callosum in experimental and control animals was performed using transmission electron microscopy (TEM) seven months after cGRP transplantation. The corpus callosum of control shiverer mice could be characterized by a lack of myelinated axons (Figure 7A). Within the brain of non-transplanted animals, nonmyelinating oligodendrocytes were visible (Figure 7A, hash), whereas in the corpus callosum of transplanted animals, myelinated axons were observed (Figure 7B,D–F, arrows) in the vicinity of functional, myelinating oligodendrocytes (Figure 7B,D–F, asterisk). A large diversity was noticed between myelination levels within the investigated region. That difference was also visible within the tissue of the same animal, dependent on the part of the corpus callosum under analysis (anterior, middle, posterior). However, if myelination was visible, its localization was most likely placed in the middle or posterior regions (Figure 7D,F), which corresponds to the IHC results. Myelin surrounding the axons of transplanted animals was significantly thicker (g-ratio = 0.7058; 0.7111 for 0.5 × 10^6^ and 0.7031 for 1 × 10^6^ transplanted animals) than in control non-transplanted animals (g-ratio = 0.8161) (Figure 7G). There was no statistically significant difference between the two transplanted groups.

### 2.4. MRI Studies of Myelination in Mouse Brains

MRI was used to capture the dynamics of cGRP-derived myelination of the dysmyelinated shiverer brain. A large amount of myelin surrounding axons within the corpus callosum influences water flow in this region and thus is visible as a hypo-intense area on T2-weighted imaging. Although in shiverer mice, due to the lack of myelin basic protein, the contrast between different brain structures in T2 scanning is not as explicit as in the case of healthy, myelinated animals, we were able to visualize the myelination process in the cGRP-transplanted mice brain. Nevertheless, the hypo-intense signal in the corpus callosum became obvious beyond 200 days regardless of the number of transplanted cells (Figure 8G) and did not change significantly after this time of observation (Figure 8A–D). No difference in the level of myelination was noticed between the animals transplanted with 0.5 × 10^6^ or 1 × 10^6^ cGRPs (Appendix A).

### 2.5. Survival Analysis

Multisite transplantation of cGRPs positively influenced host survival. The median lifespan of 0.5 × 10^6^ transplanted animals was 274 days and 261 days in the case of 1 × 10^6^ transplanted animals compared to 228 days in control non-treated mice. There was no statistically significant difference between the two transplanted groups. The comparison between the survival curves revealed a statistically significant difference (*p* = 0.0.0386; log-rank Mantel–Cox test). Moreover, in the group of experimental animals, more than 30% of cGRP-transplanted animals survived for more than 283 days and in the 1 × 10^6^ transplanted group, the lifespan of 15% of the animals was longer than 410 days (Figure 9).

## 3. Discussion

Transplantation of glial progenitors seems to constitute a promising therapeutic approach in treating demyelinating diseases. Glial progenitors may differentiate into both astrocytes and oligodendrocytes; therefore, GRP transplantation may bring beneficial effects in terms of neuronal support and myelination of axons [13,27,28,29]. In our previous study, although we saw a therapeutic effect, it largely depended on the transplantation site. Anterior transplantation (near the frontal part of the ventricles) brought better myelinating results with a less pronounced lifespan prolongation, whereas posterior transplantation (near the ventricle lining the hippocampus) resulted in lower myelination and an increase in mice survival. As we would like to achieve better results in both myelination and lifespan prolongation, we decided to multiply the cell transplantation sites (Figure 10).

In the beginning, we confirmed that cells isolated from fetal brains and cultured in vitro maintain progenitor character, presenting mostly immature antigens like Olig1, Olig2, and A2B5 antigens. Among cGRPs, only a very small fraction of cells express mature oligodendrocyte markers like O4 or the myelinating oligodendrocyte marker MBP. Cultured cells present a proliferative capacity in vitro. Thus, it seems that with the presented method of GRP isolation, we managed to obtain cells that may be characterized by typical markers representative of glial progenitors [26,30,31].

As expected, mature CC1^+^ oligodendrocytes were present in both transplanted and non-transplanted shiverer mice; however, MBP was identified only after cGRP transplantation. Like our previous results concerning cGRPs transplantation, we noticed extensive myelination in the immunohistochemical analysis [26]. The results of our studies revealed that transplanted cGRPs differentiated into functional, myelinating MBP-positive cells visible both in immunohistochemical and TEM images. What is more, we noticed actively myelinating oligodendrocytes within the tissue of transplanted animals. Moreover, the myelin surrounding axons was significantly thicker than in control non-transplanted mice (Figure 7), confirming the functional ability of the transplanted cells. We did not notice a significant increase in the number of GalC-positive cells, indicating that part of the transplanted cells differentiated into mature, myelinating cells expressing the MBP antigen. We cannot, however, exclude that the remaining transplanted GRPs that did not differentiate into MBP^+^ cells remained as a mixed population composed of immature, multipotent GRPs and their progeny, oligodendrocytes and astrocytes, as shown by others [32,33]. Similar findings with multiple injections of hGRPs in the cuprizone demyelination model were presented by Windrem and co-workers [19,34]. Interestingly, we noticed that the level of myelination was greater in the posterior parts (visible both in IHC and TEM). This fact might be related to the different number of cell infusions: three points of injection in the posterior transplantation site in comparison to two points of injection in the anterior site.

Although we used multisite injection, not all animals demonstrated meaningfully higher levels of myelination during MRI scanning, especially when compared to our previous study, where in most animals, myelination, although not prominent, was visible on MRI [26]. It needs to be stressed that although we increased the number of injections, the number of cells per point of delivery was not drastically higher. In our previous study, we transplanted 4 × 10^5^ cells per animal (2 × 10^5^ per injection). Here, we used either 0.5 × 10^6^ per animal (10^5^ per injection) or 1 × 10^6^ per animal (2 × 10^5^) per injection. In the present study, the total number of transplanted cells was increased; however, at the same time, cell dispersion throughout the brain (multiple points of injection) was also higher.

Nonetheless, such a situation might have its pros and cons: we observed the myelination process analyzing mouse brains post-mortem in IHC images and noticed that the MBP signal was dispersed throughout the whole brain. Meanwhile, we did not see dramatic changes on MRI in the same animals, probably due to the same reason—such cell dispersion means myelination in different brain regions, while MRI resolution enables us to visualize only higher myelination within the corpus callosum. Of note, the MBP marker appears relatively early during the myelination process, while changes are only visible on MRI if myelin is already relatively compact [35,36]. Therefore, it is likely that the process of myelination was stalled or that the animals were dying prior to full-blown myelination. Notably, no dose–response effect was observed. Therefore, we hypothesize that the limited therapeutic effect may depend on the insufficient trophic support of canine GRPs to the mouse dysmyelinated brain, inferior to what we previously observed in the case of human GRPs.

What is important is that multisite transplantation of cGRPs improved the survival of shiverer mice. Our previous results with a double injection of cGRPs demonstrated prolonged survival. Better results were achieved when the cells were transplanted posteriorly. Here, with the multisite infusion of cGRPs, the median survival of the Tx 0.5 × 10^6^ group was 274 days, whereas in the case of the Tx 1 × 10^6^ group, it was 261 days. In the case of anteriorly transplanted animals, only about 20% survived until the same period, and only 10% of control mice lived that long. Moreover, in the case of posteriorly transplanted animals, 40% of animals survived for 279 days and 20% for 321 days, and none of the control animals survived for longer than 223 days. In the present study, with five injection sites, in both 0.5 × 10^6^ and 1 × 10^6^ (including the posterior site), we obtained similar results to those seen in posteriorly transplanted animals in the previous study, in which about 30% of animals survived 287 days. We may speculate that the posterior injection site might be crucial in terms of lifespan prolongation. Notably, Windrem and colleagues also managed to prolong mice’s lifespan with multiple injections; however, they used human GRPs. In their case, the mice’s lifespan was greatly prolonged; however, this only occurred in 25% of the animals [19]. The authors also obtained robust myelination using anterior and posterior transplantation sites.

Similarly, Lyczek and colleagues demonstrated lifespan prolongation in the case of hGRP transplantation, with about 60% of animals surviving longer than 250 days. Interestingly, although both human and canine GRPs have a similar positive influence on lifespan prolongation and myelination of the demyelinated brain, allogenic mGRPs transplants can myelinate; however, it seems that they do not have a significant impact on lifespan prolongation [22]. Such discrepancy might be related to certain natural, species-related cell abilities to migrate, settle, and differentiate within the host tissue. Thus, for GRPs to be considered a therapeutic approach, more research is needed to understand cell limitations, especially if we consider allogeneic cell transplantation [32,37,38].

To conclude, we have proved that an increase in cell transplantation sites results in better survival of graft recipients (Table 1).

Moreover, transplanted cells differentiated into functional myelinating oligodendrocytes. However, although cGRPs and hGRPs might constitute a therapeutic perspective, transplantation with the use of cells from an allogenic source brings a lot of questions and is still to be elucidated [39]. Therefore, more research is needed to improve GRP functions, especially from the perspective of future, allogeneic therapeutic approaches.

## 4. Materials and Methods

### 4.1. Animals

All experiments were performed on shiverer mice (C3HeB/FeJ-shiverer) of both sexes (The Jackson Laboratory, Bar Harbor, ME, USA), as described previously [22,26]. Animals were bred and maintained in a temperature-controlled, specific pathogen-free (SPF) environment with a 12:12 h light/dark cycle and ad libitum access to food and water in enriched cages on wood shavings with paper nesting in groups of up to 5 in the Laboratory of Genetically Modified Animals at the Mossakowski Medical Research Institute, Polish Academy of Sciences.

### 4.2. Experimental Groups

A total of 45 shiverer mice pups (P2-3) of both sexes were used in this study. Mice were randomly assigned to one of two groups, receiving either 0.5 × 10^6^ (*n* = 15) or 1.0 × 10^6^ (*n* = 15) cGRPs (experimental group; *n* = 30) or saline (control group, *n* = 15). Throughout life, animals were subjected to magnetic resonance (MR) imaging; experimental group: *n* = 21; control group: *n* = 4. The lifespan of random animals was monitored for survival analysis; experimental group: *n* = 26; control group: *n* = 11. Random animals were also sacrificed and analyzed post-mortem by transmission electron microscopy (TEM); experimental group: *n* = 8; control group: *n* = 3; or by immunohistochemistry (IHC); experimental group: *n* = 4; control group: *n* = 4. The timeline of the animal studies is depicted in Figure 11.

### 4.3. Isolation of cGRPs and Cell Culture

Canine GRPs were isolated from mid-gestation (E32–37) dog fetuses of both sexes subjected to abortive sterilization, as described previously [26,40]. Briefly, brains were dissected and incubated in 10 mL of pre-warmed TrypLE^TM^ Express (Thermo Fisher Scientific, Waltham, MA, USA) with 10 mg/mL of DNase-1 (Sigma-Aldrich, St. Louis, MO, USA) for 10–12 min, gently triturated, and incubated at 37 °C for 10 min. Next, 5 mL of GRP medium was added [DMEM/F12 (Thermo Fisher Scientific, Waltham, MA, USA), N-2 (Thermo Fisher Scientific, Waltham, MA, USA), B-27 (Thermo Fisher Scientific, Waltham, MA, USA), 0.1% bovine serum albumin (Sigma-Aldrich, St. Louis, MO, USA), 1 µg/mL heparin (Sigma-Aldrich, St. Louis, MO, USA), penicillin-streptomycin (Thermo Fisher Scientific, Waltham, MA, USA)]. The suspension was centrifuged at 300× *g* for 5 min at room temperature (RT). The supernatant was discarded; the cell pellet was suspended in GRP medium with 10 mg/mL of DNase-1 and incubated at 37 °C for 10 min. The suspension was then centrifuged at 300× *g* for 5 min at RT, the supernatant was discarded, and the cell pellet was suspended in GRP medium supplemented with 20 ng/mL of bFGF-2 (Takara, Kusatsu, Japan). Then, the cells were plated on poly-L-lysin (PLL; Sigma-Aldrich, St. Louis, MO, USA)- and laminin (LAM; Thermo Fisher Scientific, Waltham, MA, USA)-coated T25 flasks and cultured at 37 °C in a humidified atmosphere with 5% CO_2_ until 80% confluency. At the second passage, cells were harvested, cryopreserved in ATCC freezing medium (ATCC, Manassas, VA, USA), and stored in vapor phase liquid nitrogen until further experiments.

### 4.4. Immunocytochemistry

The phenotype of cGRPs was first determined by immunocytochemistry (ICC), as described previously [26]. Briefly, cGRPs at the second passage of culture stored in liquid nitrogen were thawed and plated on PLL/LAM-coated glass coverslips. Next, the cells were fixed with 4% paraformaldehyde (PFA) for 20 min at RT, washed three times with phosphate-buffered saline (PBS), and incubated in a blocking solution (10% natural goat serum, 5% bovine serum albumin, 0.25% Triton X-100; Thermo Fisher Scientific, Waltham, MA, USA) for 60 min at RT. The following primary antibodies were used for overnight staining at the temp. 4 °C: anti-Olig1 (1:500) (AB15620, Merck KGaA, Darmstadt, Germany); anti-Olig2 (1:500) (ABN899, Merck KGaA, Darmstadt, Germany); anti-Ki67 (1:10) (PA0118, Leica Biosystems, Wetzlar, Germany); anti-PDGFR (1:200) (sc-338, Santa Cruz Biotechnology, Dallas, TX, USA); anti-A2B5 (1:200) (MAB312R, Merck KGaA, Darmstadt, Germany); anti-GFAP (1:200) (Z0334, DAKO, Jena, Germany); anti-NG2 (1:200) (AB5320, Merck KGaA, Darmstadt, Germany); anti-CNPase (1:200) (AB9342, Merck KGaA, Darmstadt, Germany); anti-O4 (1:200) (MAB345, Merck KGaA, Darmstadt, Germany); anti-GalC (1:200) (MAB342, Merck KGaA, Darmstadt, Germany); anti-MBP (1:200) (MAB382, Merck KGaA, Darmstadt, Germany). The next day, cells were washed three times in PBS and incubated with the respective fluorochrome-conjugated secondary antibodies for 60 min at RT: goat anti-chicken Alexa Fluor 488; goat anti-mouse Alexa Fluor 488; goat anti-chicken Alexa Fluor 546 or goat anti-rabbit Alexa Fluor 546 (all at 1:500) (Thermo Fisher Scientific, Waltham, MA, USA). Cell nuclei were counterstained with 5 µg/mL DAPI (Thermo Fisher Scientific, Waltham, MA, USA) for 5 min at RT. The slides were closed with Dako Fluorescence Mounting Medium (DAKO, Jena, Germany) and visualized using an Axio Observer LSM 780 microscope with ZEN Black 2012 SP5 software (Carl Zeiss, Oberkochen, Germany) at the Laboratory of Advanced Microscopy Techniques, Mossakowski Medical Research Institute, Polish Academy of Sciences. All stainings were performed in duplicate and repeated three times.

To determine the number of proliferative cells, we performed immunocytochemical staining for Ki67, as described above. Following staining, cells were photographed under 10× lens and counted manually in a blind fashion. Three regions of interest (ROIs) were selected from two separate culture plate wells. This process was repeated across three independent biological replicates. Quantification was performed using Office Excel software (MS Excel 2016, Microsoft).

### 4.5. Flow Cytometry

The phenotype of cGRPs was confirmed by flow cytometry analysis. Canine GRPs cultured from the second passage were plated on PLL/LAM-coated T25 flasks and grown until 80% confluency. Then, the cells were harvested, washed twice in PBS, centrifuged at 300× *g* for 5 min at RT, suspended in PBS + 5 mM ethylenediaminetetraacetic acid (EDTA) with 1% fetal bovine serum and the appropriate fluorochrome-conjugated antibody according to the manufacturer’s instruction (100 μL of buffer per 5 × 10^5^ cells), and incubated for 30 min at RT in the dark. The following antibodies were used: A2B5–Alexa Fluor 647 (Cat. No. 563776, BD Biosciences, Franklin Lakes, NJ, USA); O4–VioBright 515 (Cat. No. 130-120-016, Miltenyi Biotec, Bergisch Gladbach, Germany); PDGFRα–BV421 (Cat. No. 562774, BD Biosciences, Franklin Lakes, NJ, USA); NG2–PE (Cat. No. 130-123-730, Miltenyi Biotec, Bergisch Gladbach, Germany); PSA-NCAM–APC (Cat. No. 130-120-437, Miltenyi Biotec, Bergisch Gladbach, Germany), MBP–PE (Cat. No. 130-120-342, Miltenyi Biotec, Bergisch Gladbach, Germany), CD11b–APC (Cat. No. 553312, BD Biosciences, Franklin Lakes, NJ, USA) and GFAP–Alexa Fluor 647 (Cat. No. 561470, BD Biosciences, Franklin Lakes, NJ, USA). Then, the cells were washed twice in PBS + 5 mM EDTA with 1% fetal bovine serum, centrifuged at 300× *g* for 5 min at RT, and suspended in 300 μL of PBS + 5 mM EDTA. Flow cytometric analysis of cell fluorescence was performed with BD Canto II instrument with FACSDiva software v6.1.3 (BD Biosciences, Franklin Lakes, NJ, USA) with compensated parameters. Cell size (forward scatter; FSC) and granularity (side scatter; SSC) were also determined. The fluorescence intensity was measured and expressed as average for 1 × 10^4^ cells.

### 4.6. Multisite Injection of cGRPs

For transplantation, cGRPs in the second passage were thawed from liquid nitrogen, centrifuged at 300× *g* for 5 min at RT, suspended in sterile saline at a concentration of 0.5 × 10^6^ or 1 × 10^6^ per 1 μL, and stored on ice throughout the procedure. Mice pups (P2–3) were cryo-anesthetized and placed on ice in a stereotaxic apparatus with a mouse adaptor (Stoelting Co., Wood Dale, IL, USA). Each animal received five separate intracerebral injections performed using a Hamilton 1700 series syringe (Hamilton Company, Reno, NV, USA) of either (i) 0.5 × 10^6^ cGRPs; (ii) 1 × 10^6^ cGRPs; or (iii) saline (1 injection = 1 μL), using the following coordinates: AP:0 ML:1.0/−1.0, DV:0.8 from bregma; AP:0 ML:1.0/−1.0, DV:1.0 from lambda; AP:0 ML:0, DV:1.0 (lambda). After transplantation, the animals were removed from the stereotaxic apparatus, revived, and returned to their home cages. We did not notice a statistical difference in the analysis of myelination by MRI (Appendix A), g-ratio analysis on TEM images (Appendix A), or survival (Appendix A); to proceed according to the 3R principles, we decided to join two of the transplanted groups (0.5 × 10^6^ and 1 × 10^6^ transplanted cells) for IHC analysis, also taking into account the fact that myelination visible on MRI is confirmed by the presence of MBP in brain slices (IHC), which was proved previously [26].

### 4.7. Immunohistochemistry

There were 4 mice transplanted with 0.5–1.0 × 10^6^ cGRPs and 4 control mice analyzed by immunohistochemistry (IHC). The analysis was performed as described earlier [41]. Briefly, the animals were sacrificed after deep anesthesia with an intraperitoneal injection of 100 mg/kg ketamine and 1 mg/kg medetomidine, followed by transcardiac perfusion with ice-cold saline followed by ice-cold 4% PFA. Brains were dissected, post-fixed overnight in 4% paraformaldehyde (PFA), incubated in 20% saccharose until fully saturated, and cryopreserved at −80 °C until further analysis. For immunostaining, tissue samples were cryo-sectioned into 20 µm coronal sections, washed three times with PBS, and incubated in blocking solution (10% natural goat serum, 5% bovine serum albumin, 0.25% Triton X-100; Thermo Fisher Scientific, Waltham, MA, USA) for 60 min at RT. The following primary antibodies were used for overnight staining at the temp. 4 °C: anti-APC (CC1) (1:200) (ab16794, Abcam, Cambridge, UK), anti-ED1 (1:200) (MCA341R, Bio-Rad, Hercules, CA, USA), anti-β-Tubulin III (1:200) (T8578, Sigma-Aldrich, St. Louis, MO, USA); anti-GFAP (1:200) (Z0334, DAKO, Jena, Germany); anti-GalC (1:200) (MAB342, Merck KGaA, Darmstadt, Germany) and anti-MBP (1:200) (MAB382, Merck KGaA, Darmstadt, Germany). The next day, the tissue sample was washed three times in PBS and incubated with respective fluorochrome-conjugated secondary antibodies for 60 min at RT: goat anti-mouse Alexa Fluor 488; goat anti-rabbit Alexa Fluor 488; goat anti-rat Cy3 or goat anti-mouse Alexa Fluor 546 (all at 1:500) (Thermo Fisher Scientific, Waltham, MA, USA). Cell nuclei were counterstained with 5 µg/mL DAPI (Thermo Fisher Scientific, Waltham, MA, USA) for 5 min at RT. Slides were closed with Dako Fluorescence Mounting Medium (DAKO, Jena, Germany), and visualized using Cell Observer SD (Carl Zeiss, Oberkochen, Germany) microscope with ZEN Blue software v2.3 Lite (Carl Zeiss, Oberkochen, Germany) at the Laboratory of Advanced Microscopy Techniques, Mossakowski Medical Research Institute, Polish Academy of Sciences. Sections were microphotographed under 20× lens using z-stack and tile-scan commands and processed using extended depth of focus and stitching algorithms.

### 4.8. MBP Quantification after Immunohistochemical Analysis

Fixed recording settings were maintained between slices for MBP image acquisition. For each mouse, a total of 15 adjacent coronal sections were used for 3D reconstruction of the tissue cytoarchitecture (i) anteriorly (A) at the lateral ventricle depth and (ii) posteriorly (P) at the hippocampal dentate gyrus depth; chosen depths were similar between mice. Corrected total tissue fluorescence [(CTTF); CTTF = integrated density—(area of selected tissue × mean fluorescence)] [42] was used to quantify MBP across one half of the brain from each mouse, independently for the A- and P-regions. Integrated density, area, and mean fluorescence were quantified in ImageJ v1.54f (U. S. National Institutes of Health, Bethesda, MD, USA). CTTF and unpaired *t*-tests were quantified in a Microsoft Excel spreadsheet (MS Excel 2016, Microsoft, Redmond, WA, USA). * *p* < 0.05 was considered statistically significant.

### 4.9. GalC Quantification after Immunohistochemical Analysis

Fixed recording settings were maintained between slices for GalC image acquisition. From each mouse, two separate brain slices (i) encompassing the striatum or (ii) overlapping the cortex and corpus callosum were chosen for quantification. Three distinct regions of interest (ROIs) measuring 310.35 × 227.51 µm each were assigned within one-half of the brain from each mouse using ZEN Blue software v2.3 Lite (Carl Zeiss, Oberkochen, Germany). Three ROIs were assigned per each striatum and three per each cortex/corpus callosum. GalC-positive cells were quantified in ImageJ v1.54f (U. S. National Institutes of Health, Bethesda, MD, USA) within each ROI and shown as mean % ± SD of all cells (DAPI^+^). * *p* < 0.05 was considered statistically significant.

### 4.10. GFAP Quantification after Immunohistochemical Analysis

Fixed recording settings were maintained between slices for GFAP image acquisition. From each mouse, 2–4 separate brain slices encompassing the whole striatum and corpus callosum were chosen for quantification. One ROI was manually assigned within each one-half brain from each mouse using ZEN Blue software v2.3 Lite (Carl Zeiss, Oberkochen, Germany). GFAP immunoreactivity was quantified in ImageJ v1.54f (U. S. National Institutes of Health, Bethesda, MD, USA) and demonstrated as a total % of GFAP coverage within each half-brain. Results are shown as box plots with mean % ± SD denoted as whiskers. * *p* < 0.05 was considered statistically significant.

### 4.11. Ultrastructural Analysis of Mouse Brains by Transmission Electron Microscopy

The mice were anesthetized with a mixture of 100 mg/kg ketamine and 1 mg/kg medetomidine and perfused through the left ventricle, first with 0.9% NaCl in 0.01 M sodium-potassium phosphate buffer (pH 7.4), then with 2% paraformaldehyde and 2.5% glutaraldehyde in 0.1 M cacodylate buffer (pH 7.4). Isolated brain tissue samples were sequentially fixed in 2% PFA, 2.5% glutaraldehyde in cacodylate buffer, and 1% osmium tetraoxide with potassium ferricyanide for 2 h. The material was post-fixed in 1% OsO4 solution, dehydrated in an ethanol gradient, and embedded in epoxy resin Embed 812 (Serva, Heidelberg, Germany,). The ultra-thin sections were stained with uranyl acetate and lead citrate. Images were acquired using a JEM-1011 EX (Jeol, Tokyo, Japan) transmission electron microscope equipped with a MORADA camera and Analysis image processing iTEM v.1233 software (Olympus Soft Imaging Solutions, GmbH, Münster, Germany). TEM analysis was performed in the Electron Microscopy Research Unit, Mossakowski Medical Research Institute, PAS. Myelin thickness analysis was performed by measuring the g-ratio of myelinated axons exclusively. The g-ratio was measured semi-automatically using MyelTracer v1.3.1. All myelinated axons on the pictures were measured from an average of 14 FOV (where FOV is the whole TEM image). At least 23 myelinated axons were measured for one animal. Altogether, *n* = 147 (control animals) and *n* = 659 (transplanted animals) myelinated axons were measured.

### 4.12. Magnetic Resonance Imaging

MR imaging was performed at three to four time points (depending on animal survival): (i) 73–81 days, (ii) 127–130 days, (iii) 199–201 days, and (iv) 230 days after multisite cGRP-transplantation. Mice were anesthetized with 1.5–2% isoflurane/oxygen and positioned headfirst, prone in a water-heated bed. Body temperature and respiratory rate were continuously monitored throughout the experiment with MR-compatible probes (SA Instruments, Stony Brook, NY, USA). A 7T MRI scanner (BioSpec 70/30 USR, Brüker, Ettlingen, Germany) equipped with a transmitting cylindrical radiofrequency coil (8.6 cm inner diameter, Brüker, Ettlingen, Germany) and a mouse brain-dedicated receive-only array surface coil (2 × 2 elements, Brüker) were used. The protocol for structural imaging was performed as described previously [26,41]. Briefly, the T2-weighted TurboRARE sequence was used (TR = 7000 ms; TEeff = 15 ms; RARE factor = 4; NA = 4; field of view, FOV = 22 mm × 22 mm; spatial resolution = 86 µm × 86 µm × 350 µm; 42 slices, no gap; scan time ~23 min). Signal intensity measurements were performed in ImageJ v1.54f (U. S. National Institutes of Health, Bethesda, MD, USA). Briefly, the corpus callosum region was encircled manually, and its signal intensity was measured. Additionally, the intensity of two identical-in-size square-shaped regions from the left and right cortex were measured, and their mean value was treated as a background intensity. The values on the graphs represent the hypointensity of the corpus callosum related to the background (measured as a difference between background level and cc intensity). MR imaging was performed at the Small Animal Magnetic Resonance Imaging Laboratory, Mossakowski Medical Research Institute, Polish Academy of Sciences.

### 4.13. Mice Survival Analysis

To analyze the effect of cGRP multisite transplantation on mice survival, we used 38 shiverer mice. Of the 38 mice, 11 were control animals in which only saline injection was performed, and 27 animals were transplanted with 0.5–1.0 × 10^6^ cGRPs. The date of death was noted for each of the 38 animals, including the 12 transplanted animals sacrificed for tissue analysis. The date of death of the 12 sacrificed mice was treated as censored data, as the animals were taken out of this study, and we could not have followed the survival of those animals further. To compare survival curves, we used the Kaplan–Meier estimate.

### 4.14. Statistical Analysis

The differences in MBP and GalC expression between cGRP-transplanted and control shiverer mice were analyzed by the KS test. The results were shown as mean ± SD. To analyze the difference between hypointensities of the corpus callosum of positive control shiverer mice (WT, MBP^+^ animals) and transplanted shiverer mice on MR images, a one-way ANOVA was performed using Tukey’s post hoc multiple comparison test. We used the Kaplan–Meier estimate to analyze the influence of cell transplantation on mice survival. A comparison of survival curves was performed with the log-rank (Mantel–Cox) test. * *p* < 0.05 was considered statistically significant. Statistical analysis comparison between control and transplanted animals was performed using a *t*-test. * *p* < 0.05 was considered statistically significant and **** *p* < 0.0001. Statistical analysis was performed using Graph Pad Prism 7.04 (San Diego, CA, USA).

## Figures and Tables

**Figure 1 ijms-25-10580-f001:**
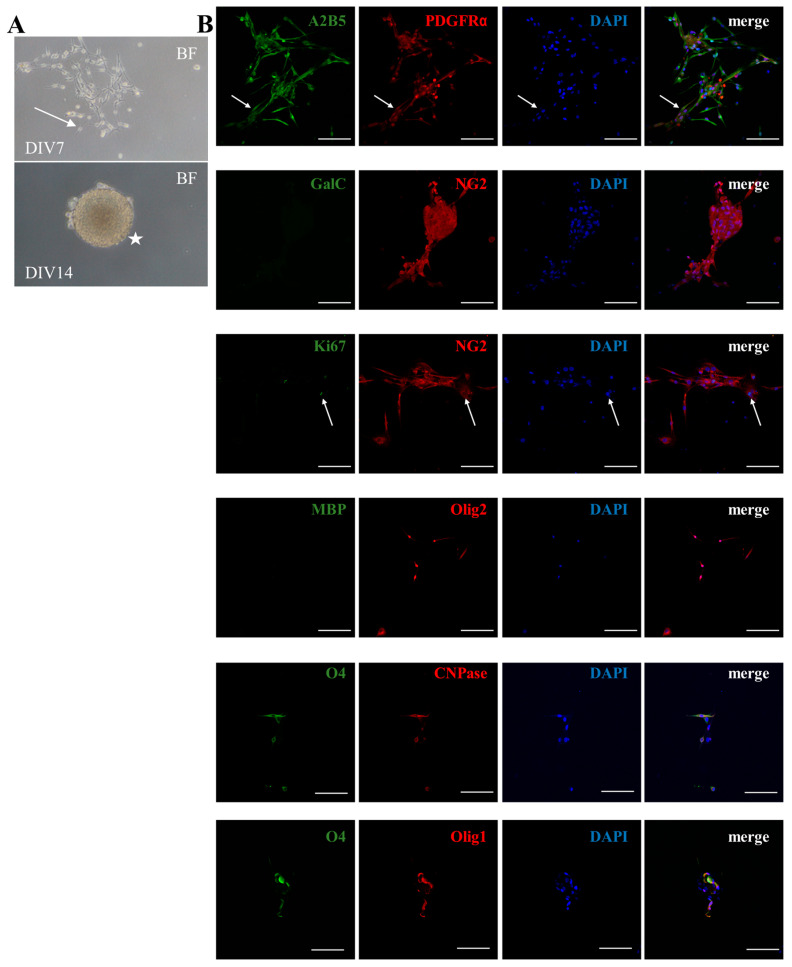
The phenotype of canine glial-restricted progenitors (cGRPs) in vitro. (**A**) Morphology of cGRPs observed during 14 days in culture. Notice bipolar morphology (arrow) and formation of cellular aggregates (symbol of a star). Abbreviations: BF, bright field; DIV, day in vitro. (**B**) Phenotypic characterization of cGRPs by immunocytochemistry. Glial progenitors/oligodendrocyte precursors were detected by expression of A2B5, NG2, PDGFRα, Olig1, Olig2, O4, and CNPase. Anti-Ki67 antibody was used as a proliferation marker and was detectable in approximately 17% of cultured cells. Anti-MBP and -GalC antibodies were used for the detection of oligodendrocytes. Nuclei were counterstained with DAPI. Scale bar: 100 μm.

**Figure 2 ijms-25-10580-f002:**
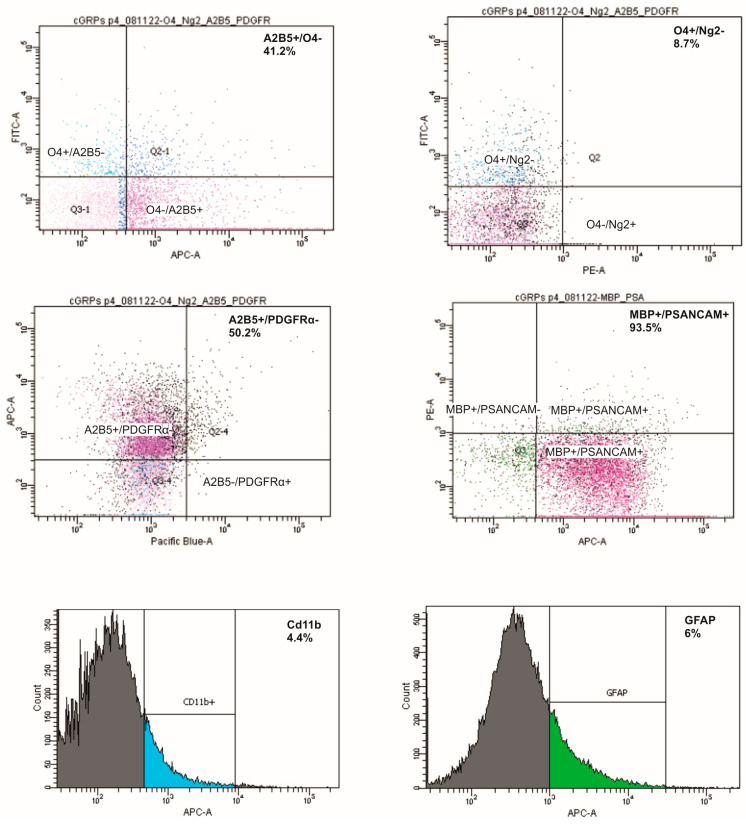
Quantitative profiling of cGRPs by flow cytometry. Most cultured cells remained in the undifferentiated, progenitor state (A2B5^+^/O4^−^: 41.2%; A2B5^+^/ PDGFRα^−^: 50.2%; O4^+^/NG2^−^: 8.7%; PSA^−^NCAM^+^/MBP^−^: 93.5%), with modest populations of GFAP+ (6%) and CD11b^+^ myeloid-lineage cells (4.4%). A2B5^−^ Alexa Fluor 647, O4 –VioBright 515; PDGFRα–BV421; NG2–PE, PSA^−^NCAM^–^APC, MBP–PE, CD11b–APC and GFAP–Alexa Fluor 647.

**Figure 3 ijms-25-10580-f003:**
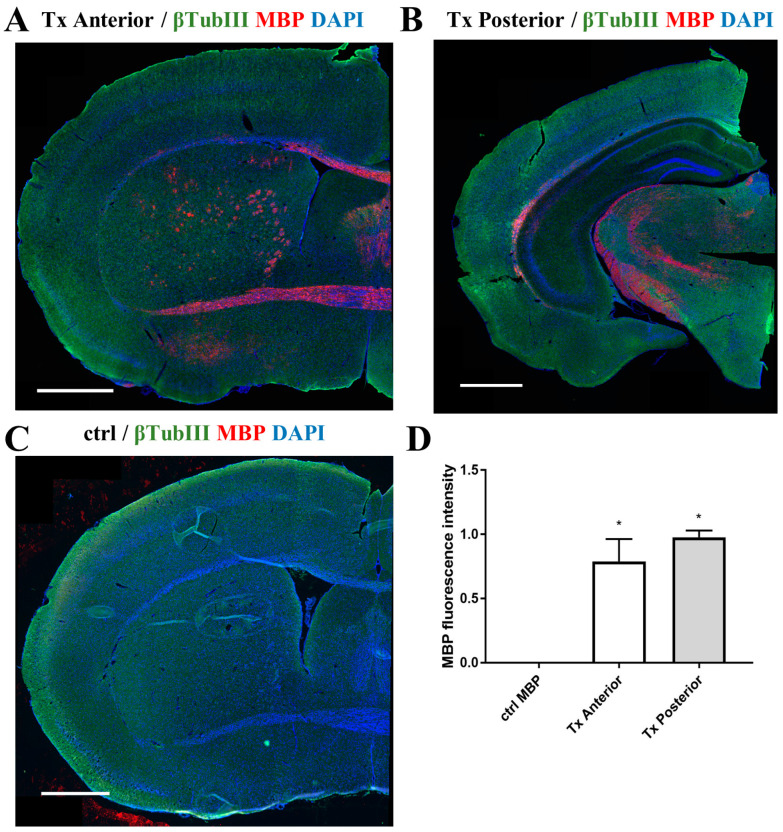
Distribution of MBP in the brain of shiverer mice 7 months after multisite cGRPs transplantation. Corrected total tissue fluorescence (CTTF) was used to quantify whole-slice MBP fluorescence. We noticed a wide presence of MBP across the brains of transplanted animals *n* = 4 (**A**,**B**), in comparison to Mbp^−/−^ control *n* = 4 (**C**). On average, 24% more MBP expression was observed in the posterior region than in the anterior brain region of transplanted animals (*p* = 0.0286) (**D**). Neurons were stained with an anti-β-TubIII antibody. Nuclei were counterstained with DAPI. Abbreviations: ctrl, control group; Tx anterior, anterior brain area from the cGRP-transplanted group; Tx posterior, posterior brain area from the cGRP-transplanted group. Scale bar: 1000 μm. Data are shown as mean ± SD. * *p* < 0.05 was considered statistically significant.

**Figure 4 ijms-25-10580-f004:**
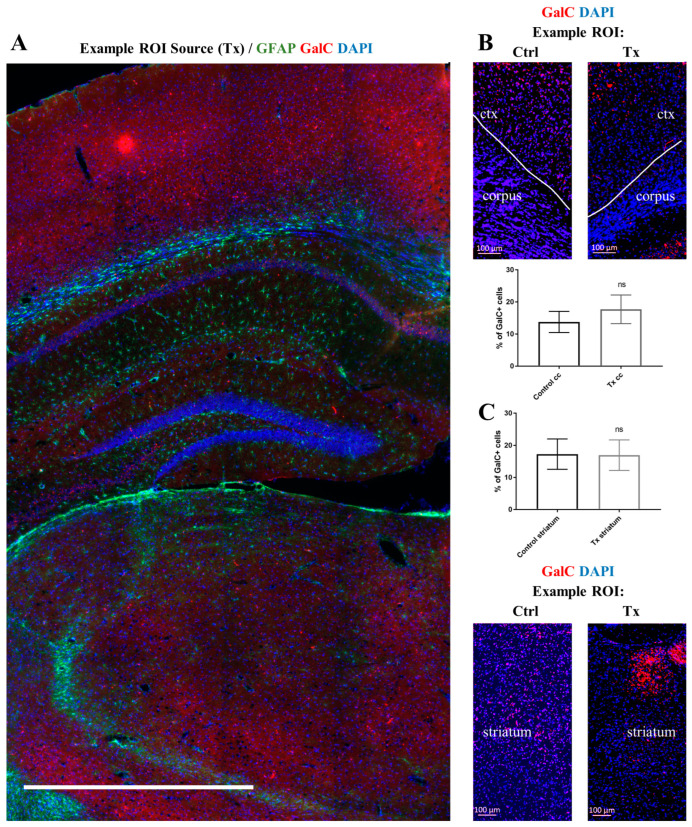
The analysis of GalC^+^ cells in the brain of shiverer mice 7 months after multisite cGRPs transplantation (**A**). The percentage of GalC^+^ cells was quantified as a mean of three separate ROIs from each cortex/corpus callosum region and three ROIs from each striatum. We did not notice any significant differences in the percentage of GalC^+^ cells between transplanted *n* = 4 [(**B**) (**Tx**) cortex/corpus callosum: 17.72 ± 1.44%; (**C**) (**Tx**) striatum: 16.98 ± 2.06%] and control *n* = 4 [(**B**) (**Ctrl**) cortex/corpus callosum: 13.79 ± 1.81%; (**C**) (**Ctrl**) striatum: 17.29 ± 2.40%)] shiverer mice (*p* > 0.05). Glial cells were stained with anti-GFAP antibody (**A**). Nuclei were counterstained with DAPI. Abbreviations: cc, cortex and corpus callosum; Ctrl, control group; ctx, cortex; ns, non-statistically significant; ROI, region of interest; Tx, cGRP-transplanted group. Scale bar: 1000 μm unless otherwise stated. Data are shown as mean ± SD.

**Figure 5 ijms-25-10580-f005:**
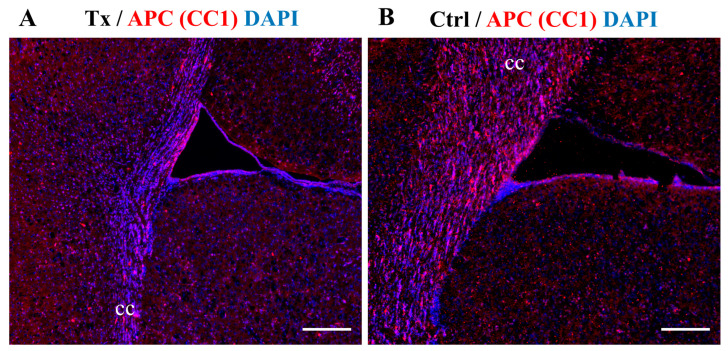
Immunohistochemical staining for oligodendrocyte bodies. APC (CC1) stainings demonstrate the presence of mature oligodendrocytes in the brain of cGRP-transplanted (*n* = 4) (**A**) and control *n* = 4 (**B**) shiverer mice. Nuclei were counterstained with DAPI. Abbreviations: Tx, cGRP-transplanted group; Ctrl, control group; cc, corpus callosum. Scale bar: 200 µm.

**Figure 6 ijms-25-10580-f006:**
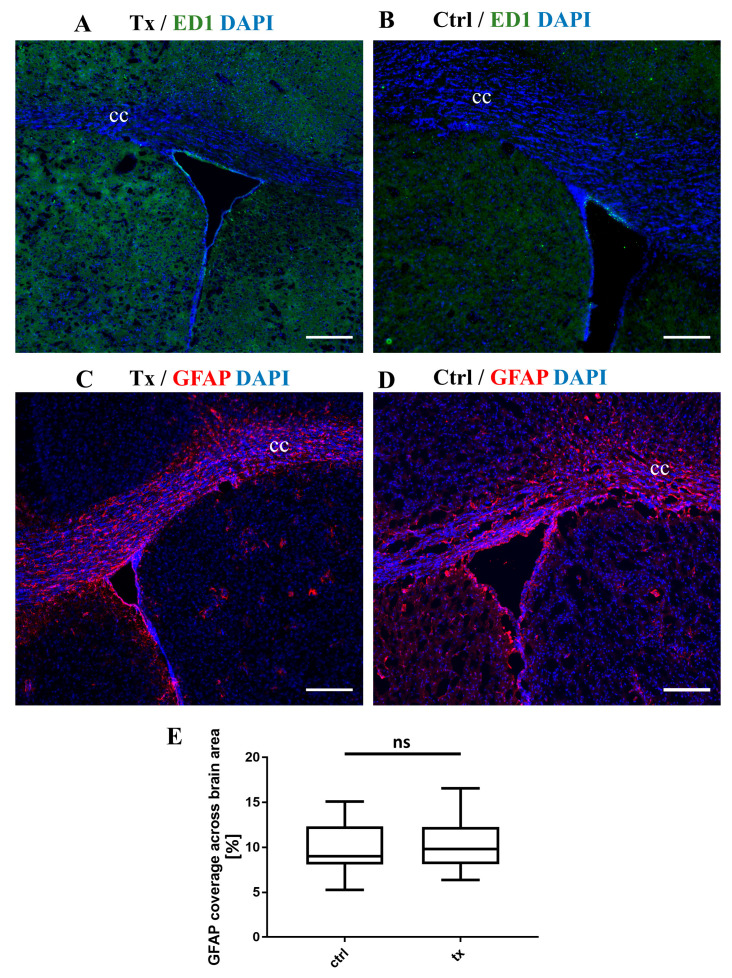
Immunohistochemical staining for activated macrophages (**A**,**B**) and reactive gliosis (**C**,**D**). ED-1 stainings demonstrated a lack of macrophage activation in the brain of both cGRP-transplanted (**A**) and control (**B**) shiverer mice. Similarly, GFAP stainings revealed no glial scar formation in the brain of cGRP-transplanted (**C**) and control (**D**) shiverer mice. Nuclei were counterstained with DAPI. (**E**) No elevation in the level of GFAP immunoreactivity was observed between the brains of transplanted (*n* = 4) and non-transplanted shiverer mice (*n* = 4). Abbreviations: Tx, cGRP-transplanted group; Ctrl, control group; cc, corpus callosum; ns, not significant. Scale bar: 200 µm.

**Figure 7 ijms-25-10580-f007:**
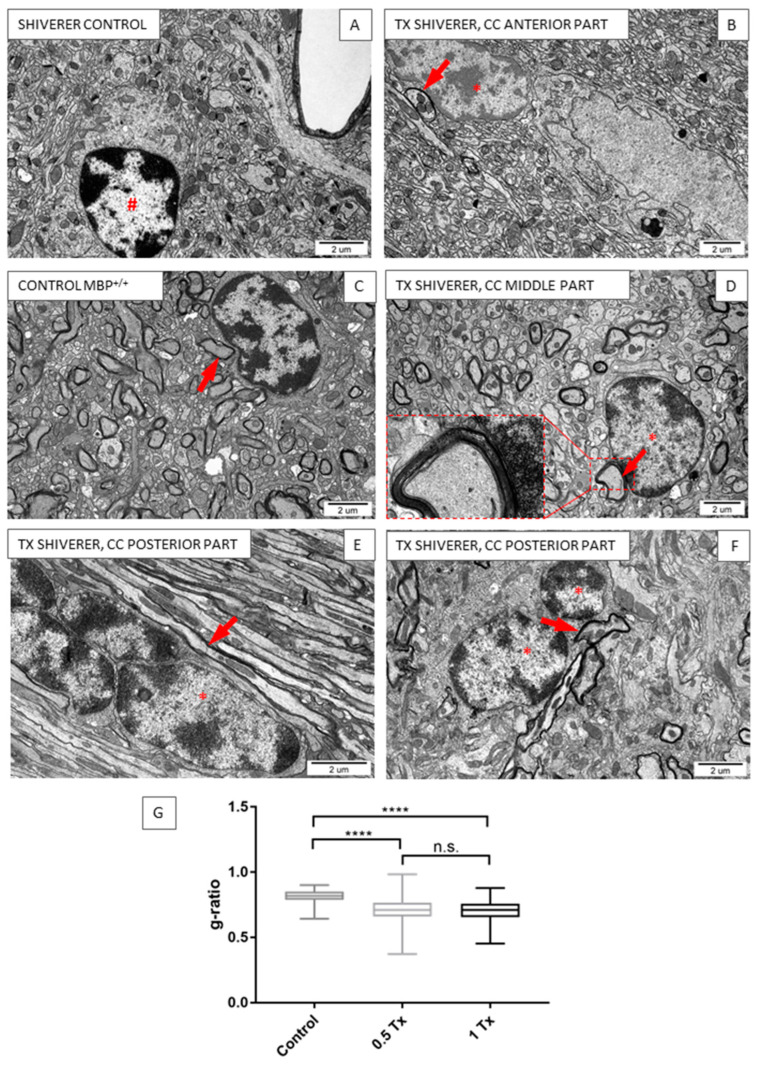
Transmission electron microscopy images presenting myelination of the corpus callosum of non-transplanted (**A**) shiverer mice compared to shiverer mice transplanted with cGRPs (**B**,**D**–**F**). The tissue of positive control mice with myelinated axons is presented in (**C**). Nonmyelinating oligodendrocytes are visible both in non-transplanted ((**A**), hash#) and transplanted animals. Myelinating oligodendrocytes located within the corpus callosum of transplanted shiverer mice ((**B**,**D**–**F**), asterisk). Arrows indicate myelinated axons. (**G**) Analysis of myelin thickness on transmission electron microscopy images presented as a g-ratio. The myelin surrounding the axons of transplanted animals *n* = 8 (tx 0.5 × 10^6^ *n* = 5, tx 1 × 10^6^ *n* = 3) is significantly thicker than myelin in non-transplanted animals (*n* = 4) as compared by g-ratio analysis. **** *p* < 0.0001, n.s.—not significant.

**Figure 8 ijms-25-10580-f008:**
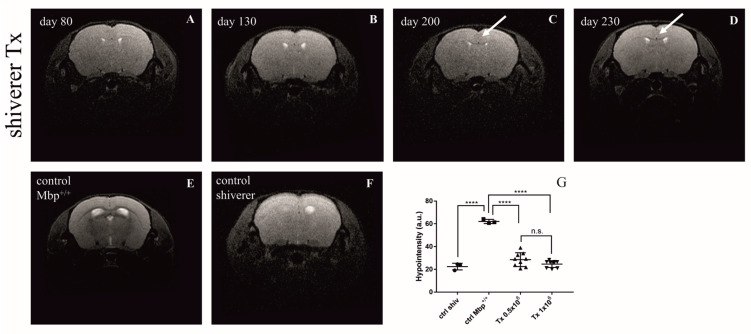
Magnetic resonance imaging (MRI) of myelination in the corpus callosum of representative: (**A**–**D**) cGRP-transplanted shiverer (Mbp^−/−^) mice at 4 time points (80, 130, 200, and 230 days post-transplantation); (**E**) control (Mbp^+/+^) mice; (**F**) non-transplanted shiverer (Mbp^−/−^) mice. Notice the appearance of myelin in transplanted shiverer mice beginning on day 200 ((**C**,**D**); arrows), extensively myelinated corpus callosum in Mbp^+/+^ mice (**E**), and undetectable myelin presence in non-transplanted shiverer mice (**F**) at the age of 200 days. (**G**) Hypomyelination level in the corpus callosum of control shiverer mice (*n* = 4), control Mbp^+^/^+^ mice (=3), and transplanted shiverer mice (tx 0.5 × 10^6^ *n* = 10, tx 1 × 10^6^ *n* = 10). No statistically significant differences were observed between transplanted groups. Abbreviations: a.u., arbitrary units; Mbp, myelin basic protein, **** *p* < 0.0001, n.s.—not significant.

**Figure 9 ijms-25-10580-f009:**
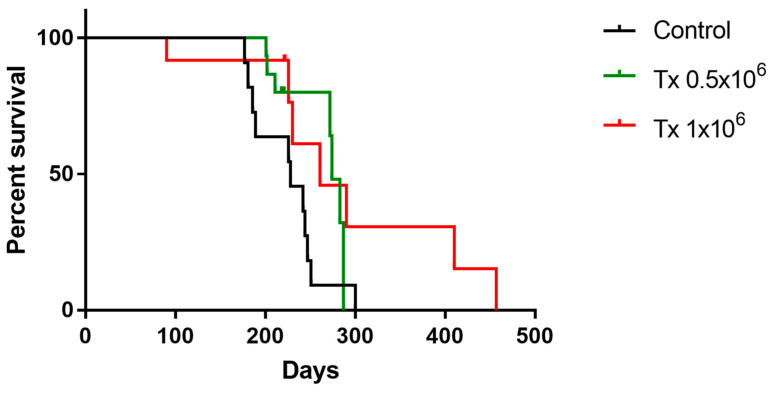
Mice lifespan analysis is depicted as a Kaplan–Meier curve. A survival curve was created for *n* = 15 of Tx 0.5 × 10^6^ and *n* = 12 Tx 1 × 10^6^ transplanted animals. Survival curve analysis reflects a lack of difference in survival of mice transplanted with different amounts of cells. Log-rank Mantel–Cox Atest *p*-value—ns., Gehan–Breslow–Wilcoxon test *p*-value—ns. The median survival of the Tx 0.5 × 10^6^ group was 274 days, whereas in the case of the Tx 1 × 10^6^ group, it was 261 days.

**Figure 10 ijms-25-10580-f010:**
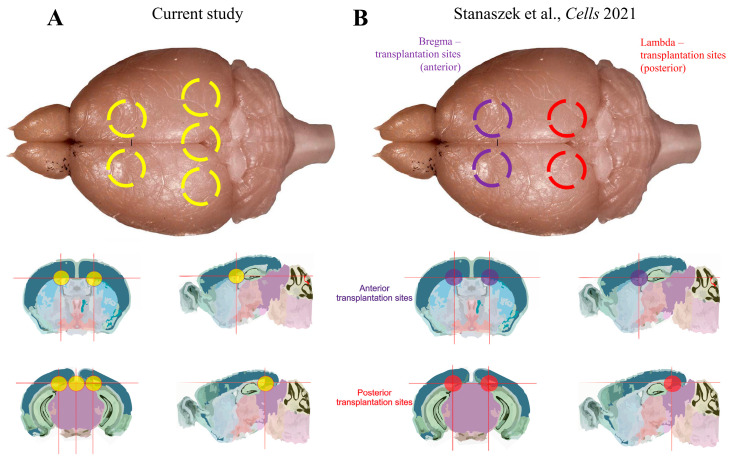
This schematic representation illustrates the cGRP injection sites in two studies: (**A**) our current study and (**B**) our previous study [26]). In the current study, we employed a multisite cGRP injection regimen, with 5 transplantation sites per mouse, represented by yellow circles. In our previous study, cGRPs were injected twice either at the bregma transplantation sites (anterior region, violet circles) or at the lambda transplantation sites (posterior region, red circles).

**Figure 11 ijms-25-10580-f011:**
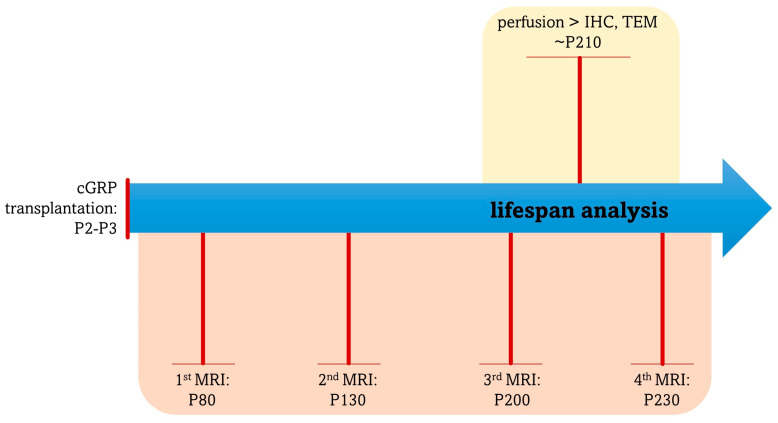
Timeline of the animal studies. Abbreviations: cGRP, canine glial-restricted progenitors; IHC, immunohistochemistry (ctrl 4, tx 0.5 × 10^6^ *n* = 2, tx 1 × 10^6^ *n* = 2); MRI, magnetic resonance imaging (ctrl *n* = 4, tx 0.5 × 10^6^ *n* = 10, tx 1 × 10^6^ *n* = 10); TEM, transmission electron microscopy (ctrl *n* = 4, tx 0.5 × 10^6^ *n* = 5, tx 1 × 10^6^ *n* = 3).

**Table 1 ijms-25-10580-t001:** Direct comparison between multisite injection (current study) and two-site injection (anterior/posterior) of cGRPs in shiverer mouse brains [26].

Aspect	Multisite Injection (Current Study)	Two-Site Injection [26]	
**Definition**	Delivery of cGRPs to multiple target sites within the brain simultaneously	Delivery of cGRPs to a single target area within the brain	
**Precision**	Simultaneous targeting of multiple brain sites	Precise targeting of a specific brain site	
**Scope**	Broader coverage across multiple brain regions	Limited to the specific site being injected	
**Impact on myelination**	High	**Anterior site:**	**Posterior site:**
		Moderate	High
**Impact on animal survival**	High	**Anterior site:**	**Posterior site:**
		Low	Moderate

## Data Availability

Data will be made available upon reasonable request.

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
