# Peer review of "Multisite Injections of Canine Glial-Restricted Progenitors Promote Brain Myelination and Extend the Survival of Dysmyelinated Mice"

_ijms, 2024, doi:10.3390/ijms251910580_

Round 1

Reviewer 1 Report (New Reviewer)

Comments and Suggestions for Authors

The authors aim to develop an optimized cGRP transplantation by using a multisite injection protocol for broader dispersal of donor cells in the dysmyelinated mice. The experiments in this article were well-designed. The animal experiments were also up to about 1 year. However, there are some problems and suggestions I suppose that the authors need to further explain and improve before this paper can be considered for publication.

1. Line 105, approximately 17% of cells revealed proliferation activity. There is no quantitative data in 1B, only fluorescent images. What method was used to get 17%? Please explain more.

2. Please write clearly in Figure 2 which antibodies are coupled with FITC and APC, and the percentage of cells in each area.

3. In result 2.2, if only immunofluorescence quantitative data is used for MBP and GalC protein quantification, it is strongly recommended to add ELISA and WB experiments for related protein quantification, which can help make the results more solid.

4. In the caption of Figure 6, line 196, how did you get ED-1 stainings that demonstrated negligible activation of macrophages? Please explain further.

Author Response

The authors aim to develop an optimized cGRP transplantation by using a multisite injection protocol for broader dispersal of donor cells in the dysmyelinated mice. The experiments in this article were well-designed. The animal experiments were also up to about 1 year. However, there are some problems and suggestions I suppose that the authors need to further explain and improve before this paper can be considered for publication.

  1. Line 105, approximately 17% of cells revealed proliferation activity. There is no quantitative data in 1B, only fluorescent images. What method was used to get 17%? Please explain more.

To determine the number of proliferative cells we performed immunocytochemical staining for Ki67, as described above. Following staining, cells were photographed under 10× lens, and counted manually in a blind fashion. Three regions of interest (ROIs) were selected from two separate culture plate wells. This process was repeated across three independent biological replicates. Quantification was performed using Office Excel software (Microsoft).

2. Please write clearly in Figure 2 which antibodies are coupled with FITC and APC, and the percentage of cells in each area.

We have added the information concerning the conjugates used in the figure 2 caption. Regarding the particular percentage in each area, we decided to insert this information in the result section, so that the figure was clear and readable.

3. In result 2.2, if only immunofluorescence quantitative data is used for MBP and GalC protein quantification, it is strongly recommended to add ELISA and WB experiments for related protein quantification, which can help make the results more solid.

We would like to thank the reviewer for the remark, however, for the purpose of this publication we cannot perform additional experiments to measure MBP and GalC by ELISA or WB. Although valuable, we do not have additional animals for the experiment. The entire experiment would took a year, keeping in mind the mice's lifespan and the experimental time points. As a confirmation of MBP, we have myelination visible both on MRI and TEM. In terms of GalC, we did not see the difference in staining in control and transplanted animals. GalC marker was used to prove that there are mature oligodendrocytes present, however without capability of myelinating. Thus, we can see that at least some of the transplanted GRPs differentiated into mature, myelinating MBP-positive cells.

4. In the caption of Figure 6, line 196, how did you get ED-1 stainings that demonstrated negligible activation of macrophages? Please explain further.

Indeed, we didn’t see much of activated ED1. However at this timepoint (218 days after transplantation) we also didn’t expect any microglial response.

Reviewer 2 Report (New Reviewer)

Comments and Suggestions for Authors

The manuscript presented by Rogujski et al entitled ‘Multisite injections of canine glial-restricted progenitors promote brain myelination and extend the survival of dysmyelinated mice’ in interesting and original, however there are several critical points to be addressed:

- The authors must improve the number of reference in the manuscript, 30 references are really a low number.

- The authors MUST perform further experiments using a specific marker for macrophage and microglia, in the results they described GFAP as a macrophage marker, but it is wrong, gfap is a general glia marker. Please to perform a set of immunohistochemistry using APOE and CD68 antibodies or other specific markers described in literature that recognize macrophage and microglia.

- The authors must explain better the experimental outline, they could insert a graph or a scheme. 

Author Response

The manuscript presented by Rogujski et al entitled ‘Multisite injections of canine glial-restricted progenitors promote brain myelination and extend the survival of dysmyelinated mice’ in interesting and original, however there are several critical points to be addressed:

- The authors must improve the number of reference in the manuscript, 30 references are really a low number.

Thank you for pointing this out. We have included additional 5 references:
[23] doi:10.1002/jnr.23332

[13] doi:10.1016/0092-8674(87)90248-0

[14] ISBN 978-0-12-648860-9

[27] doi:10.1038/s41587-023-01798-5

[33] doi:10.3390/antiox11061050

- The authors MUST perform further experiments using a specific marker for macrophage and microglia, in the results they described GFAP as a macrophage marker, but it is wrong, gfap is a general glia marker. Please to perform a set of immunohistochemistry using APOE and CD68 antibodies or other specific markers described in literature that recognize macrophage and microglia

Maybe we have missed something, or it was an editorial mistake,  but we cannot see the place where we would state that GFAP is a macrophage marker. We are conscious that this is a marker of glial cells and undifferentiated neural cells. We used GFAP as we know that GRPs are able to differentiate into both cell types oligodendrocytes and astrocytes as well as to show that there is no glial scar formation in the case of transplanted animals. At present we cannot perform additional experiments as we already are out of tissue, we didn’t expect microglia or macrophage activation at such long time points thus we performed staining for ED1 just to confirm our suspicions.

- The authors must explain better the experimental outline, they could insert a graph or a scheme

The experimental outline is shown in figure 11. We hope that it is clear enough. We might have missed adding the figure as a separate file.

Round 2

Reviewer 1 Report (New Reviewer)

Comments and Suggestions for Authors

Response to Point #1: It is recommended to put these experimental measurement data into SI

Response to Point #2: It seems that there are no changes in the figure 2 caption

Response to Point #4: It is better to have some quantitative data, such as measuring fluorescence intensity with Image J, and then conducting statistical analysis. 

It seems that there are only figure captions for the supplementary figures, but not the figures in the supplementary file. 

Author Response

Response to Point #1: It is recommended to put these experimental measurement data into SI

We have added, the experimental measurement into supllementary raw data file.

Response to Point #2: It seems that there are no changes in the figure 2 caption

Indeed, we are sorry, during pdf building we must have chosen the version after rejecting tack change. Now, everything should be added.

Response to Point #4: It is better to have some quantitative data, such as measuring fluorescence intensity with Image J, and then conducting statistical analysis.

Probably, we used the wrong term ‘negligible’. We saw some single cells that might have been activated macrophages, however, it was not a result of the injection nor injury. We did not notice the activation of macrophages within such a long time after transplantation. It might be due to two reasons: i) the cells were injected at 2 postnatal day, in that term immunological response is not yet fully developed,  ii) the transplantation procedure is not invasive as it is just a needle injection. Probably, therefore, we didn’t notice any accelerated inflammatory state in terms of reactive gliosis and activated macrophages. It might be that, we didn’t describe it correctly, and thus we have made some changes in the text stating that there is no macrophage activation. As an attachment for the reviewer, we show activated microglia/macrophages after ischemic injury (a picture made by our collaborators in the Neurorepair department) and another picture of one of our transplanted animals, where no activated macrophages were visible.

Figure 1. (A) Pictures of activated macrophages (ED1 –red) after ischemic injury.  (B) Lack of activated macrophages in cGRPs transplanted animals (218 days post-transplantation).

It seems that there are only figure captions for the supplementary figures, but not the figures in the supplementary file.

We have added supplementary figures.

Reviewer 2 Report (New Reviewer)

Comments and Suggestions for Authors

The authors must explain better the results reported in figure 6, I don’t understand, they mention macrophage activation without a specific staining using macrophage marker. In general after brain damage injection in all vertebrate there is an inflammatory response, why did the authors not considered it?

the authors must improve the number of reference, they included only 5 more reference comparing with previous version, really poor.

Author Response

Reviewer 2

The authors must explain better the results reported in figure 6, I don’t understand, they mention macrophage activation without a specific staining using macrophage marker. In general after brain damage injection in all vertebrate there is an inflammatory response, why did the authors not considered it?

We would like to thank the reviewer for the remarks. Indeed, we have used double nomenclature: in the text we have used microglia and in the figure caption macrophages. The term microglia was used previously as one cannot distinguish activated microglia from the macrophages influx from the vessels. Therefore, we are going to correct this in the text. We have used the macrophage marker – ED1 (CD68) which is commonly used for macrophages.  Indeed, we did not notice the activation of macrophages within such a long time after transplantation. It might be due to two reasons: i) the cells were injected at 2 postnatal day, in that term immunological response is not yet fully developed,  ii) the transplantation procedure is not invasive as it is just a needle injection. Probably, therefore, we didn’t notice any accelerated inflammatory state in terms of reactive gliosis and activated macrophages. It might be that, we didn’t describe it correctly, and thus we have made some changes in the text. As an attachment for the reviewer, we show activated microglia after ischemic injury (a picture made by our collaborators in the Neurorepair department) and another picture of one of our transplanted animals, where no activated macrophages were visible.

Figure 1. (A) Pictures of activated macrophages (ED1 –red) after ischemic injury.  (B) Lack of activated macrophages in cGRPs transplanted animals (218 days post-transplantation).

the authors must improve the number of reference, they included only 5 more reference comparing with previous version, really poor.

As per Reviewer’s request we have included additional 7 references, making it a total of 12 new citations comparing to the initial version:

doi:10.1002/ana.24650

doi:10.1186/s13023-021-01828-y

doi:10.1016/j.celrep.2016.06.008

doi:10.1016/j.neuroimage.2021.117744

ISBN 9789811685613

doi:10.1016/j.stemcr.2014.05.017

doi:10.1177/0963689719848355

Round 3

Reviewer 2 Report (New Reviewer)

Comments and Suggestions for Authors

the authors improved the quality of the manuscript

This manuscript is a resubmission of an earlier submission. The following is a list of the peer review reports and author responses from that submission.

Round 1

Reviewer 1 Report

Comments and Suggestions for Authors

Dear authors,

Authors demonstrated that isolated cGRPs were transplanted into multisite of brain and the cells were differentiated to myelinations oligodendrocytes in the demyelination lesion of the mice. Overall, quality of data and images in this study were pretty good. 

Reviewer's  comments as below. 

1. No idea what does authors want to demonstrate from Figure 5. Why the number of CC1-positive mature oligodendrocytes in control shiverer mice were much higher than transplanted mice? 

2. In Figure 6A and B, I believe macrophages and microglia are activated in demyelination model mice such as cuprizone-induced demyelination mice and plp1-transgenic mice. If authors refuse the reviewer's  comment,  authors have to cite supportive papers (studies).

3. In Figure 7B, an axon was myelinated in corpus callosum even though authors translated cGRPs into the region.  Why the ratio of differentiation in the region was so low compared with others. If possible, please explain it.

Comments on the Quality of English Language

Overall, the manuscript was written well in English.

Author Response

Dear authors,

Authors demonstrated that isolated cGRPs were transplanted into multisite of brain and the cells were differentiated to myelinations oligodendrocytes in the demyelination lesion of the mice. Overall, quality of data and images in this study were pretty good. 

Reviewer's  comments as below. 

  1. No idea what does authors want to demonstrate from Figure 5. Why the number of CC1-positive mature oligodendrocytes in control shiverer mice were much higher than transplanted mice? 

We did not notice a higher number of mature oligodendrocytes in control animals; actually, we aimed at a qualitative depiction. We have just stated (result section) that mature oligodendrocytes are present in control and transplanted animals, even though there are no MBP-positive cells in control shiverers. We intended to show that mature CC1+ oligodendrocytes are generally present in shiverer mice, however, newly formed myelin must be due to the transplanted cells due to malfunctioning caused by lack of MBP. We have included an additional statement in the Discussion (lines 300-302).

  1. In Figure 6A and B, I believe macrophages and microglia are activated in demyelination model mice such as cuprizone-induced demyelination mice and plp1-transgenic mice. If authors refuse the reviewer's  comment,  authors have to cite supportive papers (studies).

We have noticed only negligible activation of microglia that was comparable in shiverer controls and in shiverer transplanted animals, which is stated in the figure caption. Generally, in contrast to other models of demyelination (EAE, cuprizone) in the shiverer model there is no activation of microglia. The results with a lack of microglia activation in the shiverer model were described in the publication: https://www.ncbi.nlm.nih.gov/pmc/articles/PMC4381002/.

  1. In Figure 7B, an axon was myelinated in corpus callosum even though authors translated cGRPs into the region.  Why the ratio of differentiation in the region was so low compared with others. If possible, please explain it.

We have tried to explain it in the discussion section. Where we hypothesize this fact might be related to the localization of points of injection (see Figure 10A). 3 out of 5 points of injection are localized posteriorly, and that might be a reasonable explanation. We have also included the information in the result section that  ‘A large diversity between myelination levels within the investigated region was noticed’. The reason is that myelination visible on IHC is localized in different regions. During the animal sacrifice we are taking the probes for TEM randomly from different regions however the probes for that method are just the part of whole corpus callosum and thus we cannot be sure if we are localized in the particular area where the transplanted cells are localized (migrated to). Of course, knowing this we have taken a lot of randomly taken probes but at the same time not to overinterpret we can only speculate on the reason.

Reviewer 2 Report

Comments and Suggestions for Authors

The manuscript entitled "Multisite injections of canine glial-restricted progenitors promote brain myelination and extend survival of dysmyelinated mice" by Rogujski and colleagues aimed to optimize the therapeutic effect of cGRP transplantation, already observed in their previous studies, using a multisite injection protocol in order to achieve a wider dispersion of donor cells.

I believe that the experiments conducted are interesting and could have a positive impact on the scientific community. However, there are some points that need to be revised/better explained.

The most critical point of the whole manuscript, in my opinion, is the division into experimental groups and how the results were then described.

The authors write that they used 45 mice:

15 treated with saline (CTR);

30 treated with cGRPs (experimental group).

However: the 30 animals treated with cGRPs are actually two different groups, correct?

Where X animals (15?) received 5 injections of 10^5 cells/microliter (for a total of 0.5x10^6 cells) and the remaining Y mice (15?) received 5 injections of 2x10^5 cells/microliter (for a total of 10^6 cells). Correct?

1. The authors do not mention how many animals were treated with 0.5x10^6 cells and how many with 10^6 cells. This needs to be added.

2. If two groups of animals took different doses of cGRPs these must be considered 2 different experimental groups!! It is not possible to consider them as a single group.. Was there a difference between the two doses? Was there a dose/effect?

The entire manuscript, writing of results, and discussion should be revised on the basis of this commentary.

Minor comment:

a. In all legends, the number of animals/group and the statistical analysis used must always be indicated.

b. Figure 5: why was no quantification made? The authors are advised to include a quantification of ED1+ cells. Otherwise the image is only representative and it is not possible to state that there is no microglial activation.

c. Figure 9: The survival curve was created for CTR-mice n=11 and cGRP-mice n=27. Why?

What do the 3 missing animals/group depend on?

Note: also in this case there must be two survival curves for the two doses of cGRPs used.

d. Figure 11: is not adequate/exhaustive.

Authors must always indicate the number of animals used for each time point and analysis.

e. Tip: To increase cell dispersion and maximize the effect of CGRPs administration, did the authors try to resuspend the cells in a solution containing heparin? This could help avoid cell clustering as well as maximize dispensing after injection.

Comments on the Quality of English Language

Minor editing of English language required

Author Response

The manuscript entitled "Multisite injections of canine glial-restricted progenitors promote brain myelination and extend survival of dysmyelinated mice" by Rogujski and colleagues aimed to optimize the therapeutic effect of cGRP transplantation, already observed in their previous studies, using a multisite injection protocol in order to achieve a wider dispersion of donor cells.

I believe that the experiments conducted are interesting and could have a positive impact on the scientific community. However, there are some points that need to be revised/better explained.

The most critical point of the whole manuscript, in my opinion, is the division into experimental groups and how the results were then described.

The authors write that they used 45 mice:

15 treated with saline (CTR);

30 treated with cGRPs (experimental group).

However: the 30 animals treated with cGRPs are actually two different groups, correct?

Where X animals (15?) received 5 injections of 10^5 cells/microliter (for a total of 0.5x10^6 cells) and the remaining Y mice (15?) received 5 injections of 2x10^5 cells/microliter (for a total of 10^6 cells). Correct?

Indeed, we used two different cell numbers, 5x105 and 1x106; however, during the early MRI scans, we did not notice any difference between the groups. As mentioned in the discussion, we think it might be caused by a relatively small difference between the cell number per injection point; thus, finally, we decided to merge the groups. We also did not spot the difference, especially in other experiments (IHC and TEM). In terms of survival in both tx 105 and 2x105 groups, 50% of animals survived longer than 271 days. Till 271 days only less than 9% of control animals managed to survive. We have also checked the median survival of 3 groups (ctrl, tx 105 and tx 2x105) which were 228, 274 and 261 days. Thus also during survival analysis the difference between both transplanted groups was not visible.

  1. The authors do not mention how many animals were treated with 0.5x10^6 cells and how many with 10^6 cells. This needs to be added.

We have included the number of particular groups in the text.

  1. If two groups of animals took different doses of cGRPs these must be considered 2 different experimental groups!! It is not possible to consider them as a single group.. Was there a difference between the two doses? Was there a dose/effect?

The entire manuscript, writing of results, and discussion should be revised on the basis of this commentary.

As we did not notice the dose-effect difference in early and late MRI scans as well as TEM imaging and lastly in the IHC staining (although IHC was planned initially as a confirmation of MR imaging implementing the 3R rule to spare mice lifespan, we have expanded the analysis due to the previous reviewers' questions). As we didn’t see the difference between the transplantation groups we decided to merge them to increase the n number for IHC analysis. We have explained the possible lack of difference in the discussion section (where we claim that there is not much difference in cell number between each point of injection).

Minor comment:

  1. In all legends, the number of animals/group and the statistical analysis used must always be indicated.

For clarity, we have included additional information in all the figures.

  1. Figure 5: why was no quantification made? The authors are advised to include a quantification of ED1+ cells. Otherwise the image is only representative and it is not possible to state that there is no microglial activation.

The microglia activation was not the purpose of this work, therefore we aimed at a qualitative depiction just to confirm the lack of microglia activation. We have noticed only negligible activation of microglia that was comparable in shiverer controls and in shiverer transplanted animals, which is stated in the figure caption. Generally, in contrast to other models of demyelination (EAE, cuprizone) in the shiverer model there is no activation of microglia. The results with a lack of microglia activation in the shiverer model were described in the publication: https://www.ncbi.nlm.nih.gov/pmc/articles/PMC4381002/.

  1. Figure 9: The survival curve was created for CTR-mice n=11 and cGRP-mice n=27. Why?

What do the 3 missing animals/group depend on?

Note: also in this case there must be two survival curves for the two doses of cGRPs used.

3 mice were excluded from the survival analysis due to the death/euthanasia not related to the demyelinating condition like anus prolapse. There was no significant difference between the two groups, thus as written above we decided to merge them.

  1. Figure 11: is not adequate/exhaustive.

Authors must always indicate the number of animals used for each time point and analysis.

We have implemented the number of animals in each figure caption.

  1. Tip: To increase cell dispersion and maximize the effect of CGRPs administration, did the authors try to resuspend the cells in a solution containing heparin? This could help avoid cell clustering as well as maximize dispensing after injection.

We are thankful for the advice, we should try to implement heparin administration. Up to now, we have just used pipetting before transplantation.

Round 2

Reviewer 1 Report

Comments and Suggestions for Authors

Dear authors,

Thank you for responses to review's comments.

I think the manuscript is suitable for publication in this journal now.

Author Response

We would like to thank the reviewer for all the remarks.

Reviewer 2 Report

Comments and Suggestions for Authors

Unfortunately, I still do not consider it correct to combine two experimental groups with different dosages, without showing the data of the two groups individually.

Comments on the Quality of English Language

Pretty good, but could be improved

Author Response

We would like to thank for the reviewer’s remark. In order to respond and be clear in terms of different transplantation groups we have additionally clarified our reasoning for joining the groups. What is more, to be transparent we have included all the analyses in the supplementary figures. Therefore, the reader has the full view of what are the differences between the groups and the control.